# Survey of Keratinophilic Fungi from Feathers of Birds in Tuscany

**DOI:** 10.3390/biology10121317

**Published:** 2021-12-13

**Authors:** Simona Nardoni, Francesca Mancianti

**Affiliations:** Dipartimento di Scienze Veterinarie, Università di Pisa, 56100 Pisa, Italy; francesca.mancianti@unipi.it

**Keywords:** birds, keratinophilic fungi, dermatophytes, mycoses, fungal zoonoses

## Abstract

**Simple Summary:**

Birds represent an effective mechanism for spreading microorganisms because they can travel great distances. The present study presents the results of an extensive survey carried out to evaluate the occurrence of keratinophilic molds on several avian species from Tuscany (central Italy). These fungal species encompass several keratin-degrading fungi, frequently responsible for skin disease in both human patients and animals. Despite the fact that molds more frequently isolated belong to dark fungi, keratinophilic species, including a dermatophyte species responsible for human and animal *tinea*, were cultured in large amounts, mostly from terrestrial avian species. This finding could be due to the high water content in nests of aquatic birds, which is unsuitable for the spreading of keratinophilic molds, and due to the antidermatophytic action of the larger amounts of uropygial gland secretions produced by waterfowls when compared to terrestrial birds. Our findings confirm the important role of migrating birds in the dispersal of pathogenic and saprophytic fungi to other environments, with an evolutionarily important consequence of genetic mixing of fungal population throughout the world.

**Abstract:**

Although keratinophilic fungi on avian feathers have been widely described, data from European literature are quite lacking regarding Mediterranean countries. The aim of the present study was therefore to evaluate the occurrence of fungal species on feathers of different avian species in Italy. A total of 378 feather samples from both aquatic (n = 254) and terrestrial birds (n = 124), for a total of 30 bird species, were cultured for keratinophilic fungi. Fungal isolates were recognized by their macro- and micro-scopical morphology, and results were corroborated by PCR and sequencing. Keratinophilic fungi belonging to 11 different species (*Scopulariopsis brevicaulis*, *Chrysosporium keratinophilum*, *Trichophyton terrestre*, *Microsporum gypseum*, *Sepedonium* sp., *Chrysosporium pannorum*, *Myriodontium* sp., *Chrysosporium tropicum, Chrysosporium pruinosum, Chrysosporium luteum* and *Aphanoascus fulvescens*) were isolated from 71 animals (18.8%). The frequency of isolation of keratinophilic species from terrestrial birds was significantly higher in waterfowl. Migratory birds in Italy have been proven to carry pathogenic fungi such as dermatophytes, (*A. platyrhyncos, A. crecca, E. rubecula*), besides saprophytic species.

## 1. Introduction

Birds’ feathers are reported to play a role as a carrier of microorganisms, including fungi, which can infect other animals and humans, when predisposing factors are present [1]. The greater part of fungi described to date are saprophytes or plant pathogens. These agents are usually harmless for animals, and the few infections reported in literature as caused by them describe coincidental and noninvasive disease.

Despite this fact, there is a group of fungi able to resist hostile conditions, behaving as opportunists, such as *Cryptococcus* sp. They can survive in vertebrate tissues, and may cause deep, opportunistic mycoses in a severely immunocompromised host [2]. Furthermore, feathers represent an abundant source of keratin in the environment. For this reason, they are considered as one of the main reservoirs for keratin-degrading fungi acquired from soil, responsible for superficial skin infections in human and animals [3].

Birds represent an effective mechanism for spreading microorganisms because they can travel great distances. Some species have successfully adapted to urban environments, living in very close contact with other animal species and humans—other avian species can spread these fungi in the wild, enhancing the dispersal of these agents and the exposure of susceptible hosts to pathogens. Moreover, the handling, storage and preparing of game birds can expose human beings to keratinophilic fungi occurring on the outer contour feathers, being inhibited by body temperature of birds (about 40 °C) [4]. Most of these agents are harmless for avian hosts; *Trichophyton gallinae* is the zoophilic dermatophyte responsible for favus, usually in domestic avian hosts, able to produce specific keratinases for feathers’ keratin, and is rarely reported as cause of dermatophytosis in human and other mammals [5].

Keratinophilic fungi on avian feathers have been described mostly in terrestrial species, and a specific checklist of records in free living birds and in livestock was drafted at the beginning of this century [6]. However, data from European literature are quite lacking in relation to Mediterranean countries. To the best of our knowledge, in fact, most reports refer to non-European countries, except for investigations carried out in Northern European countries such as the early extensive basic studies on many avian species in Great Britain [7,8] and in the Czech Republic [9]. Keratinophilic flora in starlings from Brittany [1] and in chickens from Germany [10] have been reported, too.

The aim of the present study was therefore to evaluate the occurrence of fungal species on feathers of different avian species in Italy.

## 2. Materials and Methods

A total of 378 feather samples from both aquatic (n = 254) and terrestrial birds (n = 124) were examined, for a total of 30 bird species. Samples from mostly waterfowl and some from terrestrial birds were obtained from animals shot during the hunting season; in other cases feathers were collected from avian species found dead or slaughtered in avian abattoirs (*Gallus gallus domesticus*). All the animals were dermatologically healthy and did not show skin lesions. They were living in the province of Pisa, Tuscany, Italy (43°42′30″ N, 10°24′12″ E), with a Mediterranean climate, with mild winter and a hot, sunny summer season.

More detailed data are reported in Table 1.

Feathers were collected from birds and processed as previously described, with slight modification [11]. In detail, about eight feathers were plucked from different body sites of each subject and were maintained in sterile polyethylene bags until seeding. All material was cut by sterile scissors, seeded onto a selective dermatophyte medium (Mycobiotic agar, Condalab, Madrid, ES) and incubated at 25 °C until a mycotic growth was noticed (about 10 days). Subcultures of each fungal colony were achieved on Malt Extract agar. When colonies were fully developed, they were further examined, in order to micromorphologically evaluate fungal spores. Keratinophilic fungi were furtherly cultured on Potato Carrot agar, to enhance the conidia production and allow for correct identification.

Mold isolates were identified by their macro- and micro-scopical morphologic aspects, and most of the keratinophilic species were recognized at genus level. All fungi were identified following the keys provided by literature [12,13,14], yeasts were also tested for carbohydrate assimilation by ID (bioMérieux, Marcy L’Etoile, France).

The identification of keratinophilic species was confirmed by PCR. Total DNA was extracted from 20 mg of fungal growth by the Quick-DNA Fungal/Bacterial Kit (Zymo Research, Irvine, CA, USA), following the manufacturer’s instructions. A fragment of the ITS gene region was amplified by PCR [15]. Amplified PCR products were sent for Sanger sequencing to an external company. Sequences were compared against those deposited in GenBank by using the National Center for Biotechnology Information (NCBI) Basic Local Alignment Search Tool (BLAST).

A chi-squared test was used to evaluate whether significant differences exist between the frequency of isolation of keratinophilic fungi from aquatic versus terrestrial avian species.

## 3. Results

In total, 736 fungal isolates were obtained from all the examined specimens. Keratinophilic fungi belonging to 11 different species were isolated from 71 animals (18.8%). Overall, 28 waterfowls (11.7%) and 43 terrestrial avian species (30.7%) carried keratinophilic fungi, respectively.

In detail *Scopulariopsis brevicaulis* (21), *Chrysosporium keratinophilum* (20), *Trichophyton terrestre* (12,) *Microsporum gypseum* (7), *Sepedonium* sp. (3), *Chrysosporium pannorum* (3), *Myriodontium* sp. (2), *Chrysosporium tropicum*, *Chrysosporium pruinosum*, *Chrysosporium luteum* and *Aphanoascus fulvescens* (1 each), were cultured.

Data on keratinophilic fungi per avian species are reported in Table 2.

Dark fungi were also isolated. *Alternaria* spp. was the most represented fungal isolate from 181 waterfowl (76%) and 96 terrestrial birds (68.5%), followed by *Cladosporium* sp. from 129 waterfowl (54.2%) and 93 terrestrial species (66.4%). Only one isolate of *Botrytis cinerea* was obtained from *Anas crecca.*

Other environmental molds such as different species within the genera *Aspergillus*, *Penicillium*, and a few isolates of *Paecilomyces*, *Fusarium*/*Acremonium*, *Pestalotia*, *Chaetomium*, *Beauveria*, *Pochonia* and *Verticillium* were also cultured. *Rhodotorula glutinis* was the only yeast recovered from *Coturnix coturnix*, *Columba livia* and *Gallinago gallinago* (one each).

The fungal isolates were mostly associated in culture, and the number of CFU widely varied from 10 per plate to an uncountable amount, based on the texture and the aspect of different fungal species (i.e., powdery, spreading colonies of *M. gypseum*).

The frequency of isolation of keratinophilic species from terrestrial birds was significantly higher than waterfowl.

## 4. Discussion

Keratinophilic fungi were cultured from 18.8% of the selected birds, in agreement with Humplikova and Otcenasek [9], who refer to a global prevalence of 15.2% in 1120 different bird species. However, the prevalence of 30.7% found in terrestrial birds in the present study is very similar to the 36.6% and 37% reported by Pugh [7] and by Pugh and Evans [8] from birds and terrestrial avian species of Great Britain. The lower occurrence in waterfowls can be explained by a lower presence in their environment. Keratinophilic fungi are reported in fact to be more abundant in nests with a low water content, and a higher prevalence was found when the water content was below 10% of the dry weight of the nest material [8]. Furthermore, preen oil (uropygial gland secretion) has been reported to have and antifungal activity against dermatophytes [16], and waterfowl produce greater amounts of secretion than terrestrial birds (Chiale and Montalti, 2013) [17].

Data from the literature refer to a prevalence ranging from 2.2% in poultry from Germany [10] to 76.6% from terrestrial species from Antilles [3]. Differences in keratinophilic fungi depending on geographic area have been reported [6] and are confirmed by the higher prevalence found in India [18,19,20,21].

*Scopulariopsis brevicaulis* was the most frequently isolated fungal species occurring in *Gallus gallus domesticus*, being cultured from 75% of examined subjects. These findings corroborate another previous study carried out in chicken from India [22]. This mold can be involved in human onychomycoses [23] and is a rare pathogen in dogs [24].

*Ch. keratinophilum* did occur in 20 individuals. It was the most present species both in *Anas platyrhynchos* and in *Phasianus colchicus*, and has been cited in the checklist by Hubalek [6] to be cultured in conidial form or in the teleomorphic status (*Aphanoascus keratinophilum*) from these species, along with isolates from *Pica pica*. This fungal species was the more frequent *Chrysosporium* recovered from birds in Nigeria [25] and in Bahrain [11]. On the other hand, the composition of superficial oils and fats is reported to frequently affect the fungal growth, but *Ch. keratinophilum* do not appear to be significantly affected by blackbirds’ and Galliformes’ fat, in contrast with, for example, starling feathers’ fat [4]. This mycotic species is suspected to possess, together with *C. tropicum*, *Chrysosporium* state of the *arthroderma curreyi* and *Chrysosporium* state of the *arthroderma tuberculatum*, some pathogenicity [26].

*Trichophyton terrestre*, recovered from 12 specimens in the present study, has been reported to be isolated as an anamorph or in the perfect state *Arthroderma quadrifidum* from several bird species, many of which are waterfowl [9], but also from *Ph. colchicus* [6], from *G. gallus domesticus* in Japan [27,28,29] and to a lesser extent in Germany [10]. This fungal species is considered as uncommon on birds’ feathers, probably for its sensitivity to feathers’ fat and maybe to sunlight, and is reported to be recovered when it was the only fungal species [4]. However, in the present survey it was easily cultured as a sole species (from one *Ph. colchicus*), and variously associated with dark fungi, and with *S. brevicaulis* in two specimens from *G. gallus domesticus*.

*Microsporum gypseum* was the sole dermatophytic species found in birds from Tuscany, and was isolated from both waterfowl and terrestrial birds. Together with *Microsporum fulvum* and related teleomorphs, it has been isolated from *Apteryx australis*, *Alectura lathami*, Psittaciformes and Passeriformes [6]. Other records refer to domestic birds; *M. gypseum* was cultured from *G. gallus domesticus* and emu in India [19,21], and from 35% of examined domestic birds in Nigeria [25]. In our study the dermatophyte was obtained from two *Ph. colchicus* and from an *E. rubecula*, but also from four waterfowl (*A. platyrhyncos* and *A. crecca*), suggesting the suitability of feathers of aquatic birds for such geophilic mold. This dermatophyte is widely spread among animals and human beings and widely occurs in the environment, such as on hair baits shed by vertebrates (Mercer and Stewart, 2019) [30]. *M. gypseum* is frequently involved in *tinea capitis*, *tinea corporis*, kerion and several other dermatologic diseases in human patients [31,32] and in animals [33,34,35]. It has been frequently reported from animals living in cities, as well in urban soils [36]. Although other pathogenic dermatophytic species such as *Trichophyton mentagrophytes* or *Microsporum persicolor* were never found, in contrast to previous findings [9], the presence of *M. gypseum* on the feathers of seven wild birds corroborate a role of avian species as carrier of potentially pathogenic fungi.

Other fungi occasionally isolated, such as *Sepedonium* sp., *C. pannorum*, *A. fulvescens* (formerly *Anixiopsis stercoraria*), *Myriodontium* sp., *C. tropicum* and *C. pruinosum*, have been previously reported from different avian species [1,8,9,11,18,19].

*Arthroderma curreyi* is considered the fungal species most frequently isolated from feathers [8,36], especially in birds with active contact with soils [6], but it was not found in this survey. The same is true for *Ctenomyces serratus* [8].

The role of avian species as carriers of potentially pathogenic fungi has been referred to in terms of nests [37,38] and pellets [39] from northern Europe, indicating the occurrence of *Aspergillus fumigatus* and other thermophilic fungi, able to grow at 45 °C. *A. fumigatus* was in fact found in more than 50% of nests. Water content, pH and temperature during nestling affect the profile of fungal species in nests and pellets, showing the occurrence of hydrophilic, alkalitolerant and thermotolerant molds. However, in the present study, carried out in a temperate climate, *A. fumigatus* was never detected on the selected bird feathers.

## 5. Conclusions

Migratory birds in Italy are proven to carry pathogenic fungi such as dermatophytes (A. *platyrhyncos*, *A. crecca*, *E. rubecula*) and saprophytic species. This has a relevant impact not only from an epizoological point of view, but also has an evolutionary importance in terms of the genetic mixing of different fungal populations throughout the world.

## Figures and Tables

**Table 1 biology-10-01317-t001:** Avian species examined for fungal carriage on feathers and their number.

Bird Species	Total Number
*Alauda arvensis*	1
*Alectoris rufa*	3
*Anas acuta*	11
*Anas clypeata*	15
*Anas crecca*	58
*Anas penelope*	9
*Anas platyrhynchos*	63
*Anas querquedula*	4
*Anas strepera*	4
*Anser anser*	1
*Aythya ferina*	4
*Columba livia*	5
*Columba palumbus*	17
*Corvus coronae*	10
*Coturnix coturnix*	3
*Erithacus rubecula*	3
*Fulica atra*	1
*Gallinago gallinago*	44
*Gallinula chloropus*	6
*Gallus gallus domesticus*	16
*Lymnocryptes minimus*	8
*Phasianus colchicus*	44
*Pica pica*	13
*Rallus aquaticus*	4
*Scolopax rusticola*	4
*Streptopelia decaocto*	8
*Sturnus vulgaris*	6
*Turdus merula*	1
*Turdus philomelos*	5
*Vanellus vanellus*	14

**Table 2 biology-10-01317-t002:** Keratinophilic mycotic genera/species cultured from positive birds.

Bird Species	*Ch. k*	*Ch. t*	*S*	*M*	*Ch. p*	*M. g*	*T. t*	*C. pr*	*A. f*	*Ch. l*	*S. b*
*Alectoris rufa*	1	1									
*Anas clypeata **	2		1								
*Anas crecca **				2	1	2	2				1
*Anas platyrhynchos **	6					2					2
*Anas querquedula **	1										
*Columba palumbus*	1						1				1
*Corvus coronae*							1				
*Erithacus rubecula*						1					
*Gallinago gallinago **	3						2	1			
*Gallus gallus domesticus*							2				12
*Phasianus colchicus*	4		2		2	2	3		1		3
*Pica pica*	2										2
*Turdus merula*										1	

Legends. *Ch. k. Chrysosporium keratinophilum; Ch. t. Chrysosporium tropicum; S. Sepedonium sp.; M. Myrodontium sp.; Ch. p. Chrysosporium pannorum; M. g. Microsporum gypseum; T. t. Trichophyton terrestre; Ch. pr. Chrysosporium pruinosum; A. f. Aphanoascus fulvescens; Ch. l. Chrysosporium luteum; S. b. Scopulariopsis brevicaulis*. * Aquatic birds.

## Data Availability

Data used to support the findings of the present work are available from Authors upon reasonable request.

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
