# Peer review of "Survey of Keratinophilic Fungi from Feathers of Birds in Tuscany"

_biology, 2021, doi:10.3390/biology10121317_

Round 1

Reviewer 1 Report

 Paper have potential and can be published in "Biology  "Detailed remarks are below.

Birds are effective dispersers many microorganisms. Authors deal with interesting problem of   role of avian  feathers   as carrier of  fungy. Paper is very interesting and have potential  but needs some small  improvements.

  1. Format of journal should be applied everywhere.
  2. Latine names of all birds should be checked .  Plesase check especially name:  “Lymnocryptes minimus Brunnich”
  3. „Ch. keratinophilum did occur in 20 animals”. This sentemce should be check. Birds are also animals and individuals . 
  4. Could you enrich your dissusion   with the context of avian nests and pellets:  

KorniÅ‚Å‚owicz T., Kitowski I.  2013. Aspergillus fumigatus and other thermophilic fungi in nests of wetland birds. Mycopathologia 175: 43−56

KorniÅ‚Å‚owicz-Kowalska T., Kitowski I.  2009. Diversity of fungi in nests and pellets of Montagu's Harrier (Circus pygargus) from eastern Poland-importance of chemical and ecological factors.

Author Response

Reviewer 1

The Authors would thank the Reviewer for his/her valuable work, who improved the scientific value of the manuscript.

During the revision of the document, all the reviewer’s issues were answered and highlighted in yellow. They are detailed below:

  1. Format of Journal was applied everywhere
  2. Latin names were checked and modified where appropriate
  3. The sentence was checked and modified as requested
  4. The discussion section was adequately enriched

Reviewer 2 Report

The work is clearly written; every part of it is connected with logic. I don't find repetitions. It is always very important to report data on the transmission of potential pathogens to humans and other animals through other animals.

Since the methods are not described in detail, but refer to the publications on which they are based, it is clear that a consistent work of isolation and molecular identification of the fungi has been carried out.

Author Response

Reviewer 2

The Authors would thank the Reviewer for his/her valuable work

Reviewer 3 Report

The manuscript entitled “Survey of keratinophilic fungi from feathers of birds in Tuscany “ was reviewed. This is a very interesting study on a significant issue for Animal as well as Public Health.

The methodology followed is adequate for the purpose of the study and the results seem to be reliable. My main concern is the structure of the manuscript along with a few linguistic issues that should be solved by English language editing,

Some minor issues are as follows

Line 73-74: please explain which birds were slaughtered. How did you captivate them?

Line 75: It will be very interesting to add some data on the characteristics of the geographic area where the study took place. Are there any environmental conditions that could have favorited the fungal growth?

Lines 78-97: The whole procedure of fungal culture and identification is hard to follow. A flow chart could be added to better describe the steps followed.

Line 98-99: please kindly justify the use of the chi-square test.

Line 141: please refer to the journal instructions for the authors and correct the reference style

Lines 196-197: the conclusions are drawn from the findings of this study. Therefore, reference cannot be used in this part of the manuscript.

Author Response

Reviewer 3

The Authors would thank the Reviewer for his/her valuable work, who improved the scientific value of the manuscript.

During the revision of the document, all the reviewer’s issues were answered and highlighted in green. They are detailed below:

Lines 73-74: Details about the recruitment of birds were provided

Line 75: Additional data about the geographic area where the study took place were provided. We are not able to hypothesize environmental conditions favorable to fungal growth, since sampled animals were characterized by migratory habits

Lines 78-97: the procedure of fungal culture was detailed, following the Reviewer’s concerns

Lines 98-99: the chi square test was selected, being the most useful and reliable test to evaluate differences in prevalence data obtained from our research work. Anyway , the sentence was modified in order to enhance clarity

Line 141: The reference style was modified, according to the Journal style

Lines 196-197: The conclusion was modified, according to Reviewer’s observations. The reference was deleted from this part.